# Do Parents of Children with ADHD Know the Disease? Results from a Cross-Sectional Survey in Zhejiang, China

**DOI:** 10.3390/children9111775

**Published:** 2022-11-18

**Authors:** Xiaoli Fan, Ye Ma, Jingjing Cai, Guochun Zhu, Weijia Gao, Yanyi Zhang, Nannan Lin, Yanxiao Rao, Shujiong Mao, Rong Li, Rongwang Yang

**Affiliations:** 1Department of Child Psychology, Children’s Hospital, Zhejiang University School of Medicine, National Clinical Research Center for Child Health, Hangzhou 310052, China; 2Department of Pediatrics, Affiliated Hangzhou First People’s Hospital, Zhejiang University School of Medicine, Hangzhou 310006, China

**Keywords:** attention-deficit/hyperactivity disorder, knowledge, attitude, belief, parents, treatment

## Abstract

Parents’ knowledge, attitudes, and beliefs about attention-deficit/hyperactivity disorder (ADHD) are crucial in the selection of the treatment strategy and how to care for children with ADHD. However, little is known about parents’ conception in mainland China. A semi-structured questionnaire was used to assess this information with 25 true/false questions regarding ADHD, and other questions investigating the methods of acquiring ADHD-related information, treatment preference, and concerns about ADHD treatment strategy. We found that the average score of all the participants was 17.42 ± 2.69 (total of 25 points) for the questionnaire on knowledge, attitudes, and beliefs about ADHD. This indicated that the parents had insufficient knowledge of this disease profile. They always accessed specialized information through mobile internet. For the treatment options, the investigated parents chose psychotherapy treatment rather than medications, in that they worried about the side effects of medication and expected to find alternative treatment strategies. The present investigation demonstrated that most parents lack knowledge about ADHD in treatment decision making in China. Medical professionals should provide parents with evidence-based ADHD-related information to help them understand this disease.

## 1. Introduction

Attention-deficit/hyperactivity disorder (ADHD), a common childhood mental disorder, is characterized by inattention, hyperactivity, and impulsivity [1], leading to a series of functional impairments. The global prevalence of ADHD in children and adolescents is 5.29–7.2% [2,3,4], and the prevalence in China is 6.26% [5,6]. Children with ADHD face a significant increase in the incidence of school failure, suspension, disruptive behavior, lack of discipline, peer rejection, drug abuse, crime, and social dysfunction [7,8]. The updated version of the Chinese ADHD diagnosis and treatment guideline in 2015 along with the guidelines for ADHD treatment generally recommended a multimodal management approach of medications, parent training, behavioral therapy, and school interventions to improve functions [9,10,11]. These guidelines emphasize the importance of parent training in the long-term administration of ADHD and suggest that the treatment of ADHD may not be efficient without parents’ deep participation and adherence [12]. That means that the parents’ knowledge and belief play an important role in choosing the subsequent treatment strategy and good adherence to the treatment.

Studies have found that improving parents’ understanding of ADHD through training can increase compliance, which may help promote the quality of life and long-term prognosis of children with ADHD [13]. Sciutto’s research showed that lower misunderstandings about ADHD were associated with higher medication acceptance [14]. Research indicated that the contents of psychoeducation in parent training are the key to benefitting the process of ADHD treatment. There are a few studies showing that parents’ awareness of ADHD is not sufficient [3,7,15,16,17]; however, they were performed outside of China. Little is known about parents’ knowledge, attitudes, and beliefs about ADHD in mainland China. In addition, parents’ expectations and concerns about treatment measures and the exact information needed in clinical practice have not yet been investigated.

Therefore, the present study aimed to (a) investigate parents who have children with ADHD regarding their knowledge, attitudes, and beliefs about this disorder, (b) identify the methods of acquiring information about ADHD, and (c) learn parents’ preferences, concerns, and expectations regarding ADHD treatment. Our findings reveal parents’ understanding of ADHD in Zhejiang Province, China, and corresponding educational materials should be provided to these parents in need.

## 2. Materials and Methods

This was a cross-sectional, descriptive study. The recruited participants, consisting of only one of the parents having a child diagnosed with ADHD for the first time based on DSM-5 criteria, were interviewed with a revised version of the questionnaire, including demographic characteristics, parental knowledge, attitudes, beliefs, treatment goals and concerns, treatment selection preference, and any related content that parents greatly wanted to know. The contents of parental knowledge, attitudes, and beliefs in the questionnaire were quoted and adapted from the research questionnaires made by Mulholland [18] and studies by Ghanizadeh [3,19], which were shown to have good internal consistency and repeated-measure consistency. The other parts were designed by the authors with respect to the clinical needs to collect descriptive information. A pilot study was carried out on 20 parents, and survey questions were revised according to the results. The Chinese version of the questionnaire is available from the authors upon request.

### 2.1. Sample

The sample consisted of the parents of children with ADHD aged 8 to 16 years. These children were referred to the Children’s Hospital, Zhejiang University School of Medicine, from August 2020 to September 2021. Each primary caregiver of the individual referred to the clinics was included in the study if the child met the inclusion criteria and provided consent. During the period, it was emphasized that the information collected would be confidential and would only be used for analysis to improve the psychoeducation of parent training in ADHD management.

### 2.2. Questionnaire

The first part collected demographic characteristics, including child gender and age, parent gender, age, and education level. The second part included a 25-item survey that was used to collect data on the knowledge, attitudes, and beliefs of the parents of children with ADHD. It consisted of 25 true/false questions (wrong answer = 0; right answer = 1). The third part included three multiple-choice questions that were used to collect the methods of acquiring ADHD-related information, treatment preferences, and concerns about ADHD treatment strategy. The fourth part included two open-ended questions that were used to collect the treatment goals and the relevant information they most wanted to know.

### 2.3. Sample Size

This study was a cross-sectional and experimental study. The purpose was to investigate the level of ADHD-related knowledge among the parents of children with ADHD in China. A two-sided test was performed, and the test criterion was set as α = 0.05. The mean ADHD-related knowledge score was 17.15, and the standard deviation was 3.10 in the pilot study. The admissible error was 0.43. The sample size was calculated using the PASS software version 15 (n = 201). Considering that 40% of the participants did not completely finish the questionnaire, the target enrollment of 281 patients was achieved in this study.

### 2.4. Data Analysis

The variables were input into SPSS 25 for statistical analysis. Descriptive analysis was used to analyze the demographic characteristics, and a chi-square test was used to analyze whether there were differences in the age and educational level of the fathers and mothers. Meanwhile, the distribution of parents’ knowledge, attitudes, and beliefs about ADHD, the percentage of parents who accessed ADHD-related information from different sources, their views on ADHD treatment, and their concerns were descriptively analyzed. At the same time, a chi-square test and Wilcoxon rank-sum test were used to analyze the difference between the mothers and fathers.

For all the analyses, *p* values less than a level of significance value of 0.05 were considered statistically significant.

## 3. Results

### 3.1. Participants’ Characteristics

A sample of 295 parents of children (n = 295) with ADHD were potentially included in the study. Finally, 210 were recruited. Three-quarters of them were mothers (N = 154). One-half of the parents had an education level of college or above (50% of fathers and 53.9% of mothers). There was no difference between the age and education level of the fathers and mothers ($\chi^{2}$ = 4.884, *p* = 0.180). The main demographic characteristics are shown in Table 1.

### 3.2. Parents’ Knowledge, Attitudes, and Beliefs about ADHD

The average score of all the participants was 17.42 ± 2.69 (95% CI: 17.06–17.78) (total of 25 points) for the questionnaire on knowledge, attitudes, and beliefs about ADHD. The scores from the mothers were higher than those from the fathers, but the difference did not reach statistical significance (17.34 ± 3.08 vs. 17.45 ± 2.54, Z = −0.190, *p* = 0.849). There was no statistical significance among parents with different educational levels regarding their knowledge, attitudes, and beliefs about ADHD (F = 1.618, Z = 0.201)

Regarding the nature of ADHD, most parents believed ADHD was a real disease (60.95%), not a manifestation of a child’s curiosity (87.14%), and could not heal itself (81.90%). However, approximately one-third of the parents considered ADHD to be a bad habit and only occurred in childhood.

Unfortunately, regarding the topic of the etiology of ADHD, 40.48% of the parents considered ADHD to be caused by sugar intake and food additives, and 70.48% of the parents believed that the wrong method of parenting would be a reason for this disease (Table 2).

In terms of functional impairments, more than three-quarters of the parents thought ADHD lowered the functions of academic achievements (80.95%), peer relationships (84.29%), and interaction with family members (78.57%). More than two-thirds of them knew that children with ADHD were at severe risk of being disciplinary violations (83.81%) and playing truants (73.90%).

As regards the treatment strategy, 78.10% of the parents believed ADHD was mainly dependent on drug treatment, and 93.33% believed that children should be parented under the same rules as their normal peers. A total of 82.86% of the parents took the view that excessive punishment increased behavioral problems in ADHD children. Almost all the parents thought children with ADHD needed psychological support, and teachers should understand ADHD and coping methods.

Regarding the item of homework, only 27.14% of the parents thought students with ADHD should have less homework, and 42.38% thought they should have more oral homework. However, more than one-third (37.14%) of the parents thought that children with ADHD were more suitable for extra course tutoring. There was no difference between fathers’ and mothers’ views about all the questions in the questionnaire except item 21 ($\chi^{2}$ = 5.941, *p* = 0.015), which is shown in Table 2.

### 3.3. The Methods of Parents Acquiring Information about ADHD

The manner of acquiring information about ADHD for most parents was mobile internet using smartphones, from ”moments” or official institute accounts in WeChat or other media. They rarely used professional books or academic websites to find ADHD-related resources. Other information is shown in Figure 1.

### 3.4. Parents’ Preferences, Concerns, and Expectations Regarding the Outcomes of ADHD Treatment

Overall, 97% of the parents believed that their children needed psychological support or behavioral therapy, and they preferred the treatment of psychological support treatment (60.0%) for their children, followed by behavioral therapy (43.33%), or by medications (28.10%). Approximately half of the parents were concerned about the side effects of treatment. One-third of the parents expected their children’s attention, behavior, and learning to improve after treatment. In addition, 28% of the parents wanted to learn about the treatment methods or side effects of medications before beginning the treatment. More detailed information is shown in Table 3.

## 4. Discussion

Parents’ perceptions about ADHD are crucial in how to select the next intervention strategy after diagnosis, which greatly influence the long-term treatment outcomes [7,20]. Studies have shown that low levels of parental knowledge mediate the association between ADHD symptoms and risk-taking behavior [21] and other domains of impairments [22]. Bennett et al. found that parents’ ADHD knowledge was positively related to their medication acceptability and correct treatment strategy [16]. The long-term outcomes of ADHD were improved with a consistent and standardized treatment model of medication and behavior therapy [23]. Therefore, parents’ knowledge and beliefs are key to the prognosis of ADHD. There have also been a series of field studies in the US, the Netherlands, the UK, and Africa [14,17,22,24,25,26]. The present investigation was one of the few studies on the topic of parents’ knowledge, attitudes, and beliefs toward ADHD in mainland China.

This study revealed that the knowledge and beliefs about ADHD were insufficient (overall correct rate 68.6%) in parents who had a child with ADHD, these children were diagnosed as ADHD for the first time. These results were consistent with the findings of Amiri et al. [27], which showed that 66% of parents had general ADHD knowledge in Iran. In addition, our results were also in agreement with another study in South Africa [28]. That study was a cross-sectional survey of 79 South African parents with ADHD children about the knowledge of ADHD, indicating that their awareness of ADHD symptoms or diagnosis accounted for two-thirds of the total score; treatment awareness accounted for half of the total score, and the related characteristics (i.e., general knowledge about nature, cause, and outcome) accounted for only a quarter. That meant parents had limited awareness of ADHD in South Africa [28]. Our results, combined with the results of previous studies, indicate that there is a long way to go for parents to educate themselves about ADHD with proven evidence. There was no statistical significance among parents with different educational levels regarding their knowledge, attitudes, and beliefs about ADHD, which was consistent with other studies [27,28,29]. However, the family environment (parents’ education level, family socio-economic status, family composition and environment, parenting behavior and interaction methods, parents’ mental health and function, parents’ material use, etc.) is closely related to the psychological condition of children and externalizing disorders [17,30]. This indicates that the level of education of parents has no relationship with the level of knowledge and understanding of parents but is related to the prognosis of the disease. In addition, it is worth noting that some of the experiences of parents, including trauma, hunger, poverty, etc., could also increase the risk of mental disorders and their related diseases in the next generation [31,32]. When exploring the related causes of ADHD and the education of children’s families, the experience and psychological conditions of parents cannot be ignored.

Our survey indicated that many participants had incorrect beliefs about the nature of ADHD. Approximately one-third of the parents considered ADHD as a bad habit and considered ADHD to occur only in childhood, which was also drawn in one study in Iran [29]. With these wrong points of view, parents would prefer to punish children with ADHD or just wait for it to outgrow. Children with ADHD are more likely to receive criticism and punishment [33]. When punishment is used to manage and shape the behaviors of children with ADHD, it may bring them unexpected and unnecessary side outcomes [34], such as negative parent–child relationships and more severe functional impairments [35]. This is a neurodevelopmental disorder, and 10%–60% of children and adolescents with ADHD have symptoms persisting into adulthood [2,9,16,36], which affects many aspects of the patient’s life, including their physical health, mental health, society, occupation, school and family relationships [2,16]. Parents’ correct knowledge of the nature of the disease would lead to more understanding and tolerance for children with ADHD, which would benefit the outcomes.

We also found that most parents had misunderstandings about the causes of ADHD. It is worth noting that two-thirds of the parents believed that the wrong method of education had caused this disease, which was consistent with the research by Ghanizadeh et al. [3]. Attributing ADHD to the wrong way of parenting behaviors might cause parents to feel guilty and have compensating thoughts for children, which would contribute to excessive doting and the degeneration of children’s behaviors [3]. In a focus interview survey in America, 43% of the parents believed that ADHD was caused by too much sugar in the diet, while 14% of the parents did not know whether sugar played a role in the cause of ADHD [37]. Our study found that nearly one-third of the parents believed sugar intake causes ADHD, which was higher than previously mentioned studies. The viewpoint that ADHD was related to sugar intake and dietary intake was popular among the surveyed participants. Johnson et al. [38] found that excessive sugar intake leads to an increase in dopamine release and a decrease in dopamine D2 receptors, which leads to a decrease in the response of dopamine to sugar over time and eventually lowers the sensitivity of the frontal lobe to reward decreases, leading to the continued development of ADHD. However, the relationship between sugar and ADHD has not been confirmed. DiBattista et al. reported that teachers mislead parents in terms of sugar intake and ADHD [37], so our participants’ thought that sugar causes ADHD was more likely to be some old beliefs. If it was correct, sugar restriction would be the main treatment strategy rather than medications. Parental training programs should focus on the etiology of ADHD in the future, which would benefit parents’ treatment preferences.

Most parents believed that teachers should be aware of ADHD and know the management in this study. The teachers of children with ADHD are important in the process of evaluation, treatment, and follow-up of the disease. One interview with parents with ADHD children about ADHD management also found that those parents frequently raised the point that school and teacher support was necessary for ADHD management in America [39]. A randomized study [40] on the evaluation and treatment of ADHD showed that pediatricians relied on the relevant information reported by teachers when diagnosing and treating children with ADHD. One meta-analysis [34] showed that behavioral parent and teacher training could reduce opposing behaviors, improve positive education, reduce negative and strict parenting, and increase parents’ sense of nurturing ability. These results showed that most parents recognized that teachers were important in the treatment of children with ADHD. However, it is unknown whether teachers had learned the knowledge of ADHD, which should be explored in the future.

There are several functional impairments of ADHD, including low academic performance, violation of discipline, poor peer relationships, and emotional dysregulation. Appropriate requirements for them could benefit the outcomes. However, our study found that few parents thought that students with ADHD should have less homework and needed more oral homework. This showed that parents usually do not know that lowering the quantity of homework for children with ADHD is better, nor do they know which way of learning is more suitable for ADHD students. Emotional dysregulation and poor peer relationships should be addressed, as they would exacerbate the symptoms of ADHD and contribute to more severe functional impairments of ADHD [33]. More than three-quarters believed that children with ADHD have difficulties in peer relationships and family interactions. Consistent with the results from studies in America and Iran [29], the awareness of most parents regarding these aspects would help them choose a treatment combining pharmacological therapy with psychotherapy. Fortunately, in our survey, most parents knew that children with ADHD were at serious risk of experiencing disciplinary violations and playing truants.

In total, the knowledge, attitudes, and beliefs toward ADHD were similar between fathers and mothers in the present study. However, the rate of the included mothers was higher than that of the fathers, which was consistent with a study in Mexico [24,41] and meant that mothers are always the primary caregivers of the children. The caregivers of children with ADHD experience great pressure [42]. From this point of view, the mental health status of those mothers who have children with ADHD should receive more attention and acquire more support.

With respect to the treatment strategy, although three-quarters of the parents believed ADHD was mainly dependent on pharmacological treatment, the first treatment option was psychological support, followed by behavioral modification and medications. However, there were more parents who believe that medications can control ADHD in Iran [29]. Most guidelines recommend medications, such as stimulants and atomoxetine combined with behavioral therapy, as the first-line treatment for ADHD [11,15,43], and these effects were also proven in RCTs. The current first-line medication for the treatment of ADHD is psychostimulants (methylphenidate and amphetamines). The common side effects of such drugs include the loss of appetite and insomnia. In fact, a low rate of ADHD medication use or persistent use was a problem in America, Australia, and China [25]. Our research found that about half of the parents of children with ADHD were worried about drug side effects. However, it is not easy to persuade the parents of children with ADHD to take medication as the first option. More and more studies are exploring the mechanism of the activity of ADHD therapeutic drugs. Research on these mechanisms helps guide the development of more refined and tolerable treatments, thereby improving medical compliance and long-term efficacy [44]. Our results indicated that the parents knew that medications could benefit ADHD symptoms, but they preferred psychological therapy. Generally, behavioral modification is recommended only for children of younger age and with mild ADHD symptoms [2]. For those with severe symptoms and aged over 12 years, the monotherapy of behavioral modifications may not work well, and psychopharmacological drugs are indispensable [11]. Overall, parents’ knowledge of the treatment of this disease is insufficient.

The methods of acquiring information about ADHD for most parents were via the internet using smartphones. They rarely used professional books or academic websites as resources for ADHD knowledge. Similar results were also found in other studies in South Africa and America [7,28]. With the rapid development of science and technology, the internet has become an important and convenient medium for facilitating access to and dissemination of information [7,45], but studies have found that the ADHD-related information obtained on the internet is usually unprofessional, massive, unorganized, and confusing [46,47]. The dissemination of inaccurate and outdated information and fragmented knowledge other than systematic knowledge is always contradictory, which confuses the parents in drawing a conclusion [48], influences their decision on how to treat their children [7,47,48,49], and even raises questions [7]. Most parents have no access to the right information. This is one of the reasons why parents and professionals have inconsistent attitudes toward medication treatment. Therefore, it is very important for professionals to provide parents with reputable websites or WeChat official accounts, and parents should also take the initiative to learn the correct relevant information through these resources.

As we hypothesized, the main content that parents wanted to know was the treatment-related information that was basically regarding the parents’ concerns. In one study in America, parents worried about side effects and long-term adverse effects and thought that medication should be their last resort [20]. Another study [15] indicated that parents and children with ADHD are unlikely to have good compliance without the opportunity to discuss their concerns with their doctors. Increasing parents’ understanding of treatment could improve their adherence to treatment [50].

This study had some limitations. Our cross-sectional study could not determine whether improving parents’ knowledge, attitudes, and beliefs about ADHD could improve patients’ outcomes, which can be investigated in a future longitudinal cohort. The participants in this study were the parents of children with ADHD, and the results of the study might not be representative of the views of other non-diagnosed ADHD patients. Moreover, there may be significant differences between the parents of these newly diagnosed children with ADHD and those already in the treatment and management stages of the disorder. Of course, our study focused on the parents of children with ADHD diagnosed for the first time. Therefore, the parents of children with ADHD have limited views on ADHD. Our sample size was also limited, which might result in bias and could not provide a more convincing conclusion. In addition, the perceptual credibility obtained by various information sources was not effectively detected; this information helps to optimize the priority information sources of psychological intervention education resources, but this study can be used as a starting point and provide some relevant data in mainland China.

## 5. Conclusions

Our study revealed that a large gap still exists between evidence-based knowledge and the opinions of parents with children diagnosed with ADHD for the first time. Quite a few parents in the present research had an insufficient understanding of the symptoms, etiology, nature, treatment, and prognosis of ADHD. Parents preferred psychotherapy and wanted to know more information on treatment. Therefore, medical professionals should provide parents with evidence-based ADHD-related information in parent training and establish official WeChat accounts or reputable websites to answer the concerns of parents. The present results should be taken into account when planning education materials for parents with children with ADHD.

## Figures and Tables

**Figure 1 children-09-01775-f001:**
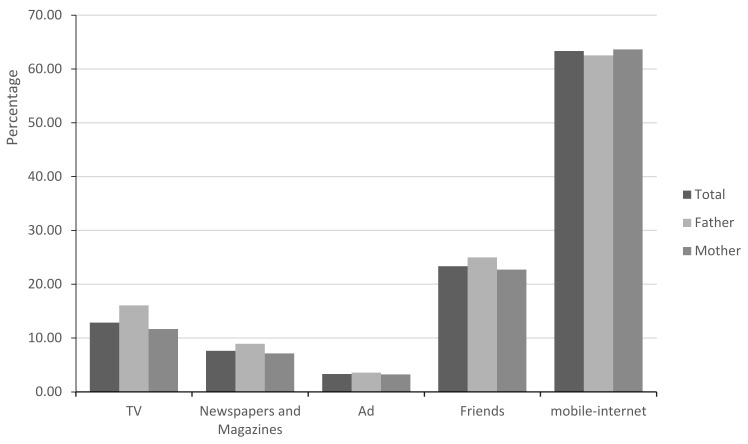
Sources of parents for accessing information.

**Table 1 children-09-01775-t001:** The characteristics of parents with children affected by ADHD.

Items	N (%) of Fathers	N (%) of Mothers	Chi-Square Value	*p* Value
Age group of participants (y)			4.884	0.180
26–30	1 (1.79)	8 (5.19)		
31–35	17 (30.36)	65 (42.20)		
36–40	27 48.21)	53 (34.42)		
>40	11 (19.64)	28 (18.18)		
Participant’s educational level			0.872	0.647
Less than high school	11 (19.64)	34 (22.08)		
High school	17 (30.36)	37 (24.03)		
College and advanced degree	28 (50.00)	83 (53.90)		
Total (N = 210)	56 (26.67)	154 (73.33)		

**Table 2 children-09-01775-t002:** Detailed percentage of correct answers in response to questions on knowledge, attitudes, and beliefs about ADHD (N (%)).

Item Number	Statements of Knowledge, Attitudes, and Beliefs	The Statement Is True (T) or False (F)	Correct Answer of All Participants (n, %)	Correct Answer of Fathers (n, %)	Correct Answer of Mothers (n, %)
1	ADHD is an illness that is linked to biological and genetic issues.	T	128 (60.95)	33 (58.92)	95 (61.69)
2	ADHD is an illness occurring only in childhood.	F	145 (69.05)	37 (66.07)	108 (70.13)
3	ADHD is simply a manifestation of a child’s curiosity.	F	183 (87.14)	48 (85.71)	135 (87.66)
4	ADHD is a serious issue that affects children’s health.	T	191 (90.95)	52 (92.86)	139 (90.26)
5	ADHD is an illness that heals itself.	F	172 (81.90)	42 (75.00)	130 (84.42)
6	ADHD may be due to the parents’ wrong way of education and doting.	F	62 (29.52)	22 (39.29)	40 (25.97)
7	Children with ADHD are at serious risk of being absent and playing truant.	T	151 (71.90)	39 (69.64)	112 (72.73)
8	ADHD has negative effects on life.	T	199 (94.76)	54 (96.43)	145 (94.16)
9	Supplementary course tutoring is more appropriate for children with ADHD.	F	132 (62.86)	32 (57.14)	100 (64.94)
10	Children with ADHD are at greater risk of disciplinary violations.	T	176 (83.81)	48 (85.71)	128 (83.12)
11	Children with ADHD have a higher IQ than non-ADHD children	F	170 (80.95)	43 (76.79)	127 (82.47)
12	Children with ADHD need psychological support.	T	206 (98.10)	54 (96.43)	152 (98.70)
13	Children with ADHD should be subjected to discipline and rules as other children.	F	14 (6.67)	1 (1.79)	13 (8.44)
14	Teachers should understand ADHD and coping methods.	T	197 (93.81)	54 (96.43)	143 (92.86)
15	Children with ADHD have more difficulties in interacting with peers.	T	177 (84.29)	44 (78.57)	133 (86.36)
16	Children with ADHD have worse academic achievement than non-ADHD children.	T	170 (80.95)	44 (78.57)	126 (81.82)
17	Children with ADHD have more difficulties in interacting with family members.	T	165 (78.57)	43 (76.79)	122 (79.22)
18	Students with ADHD should do less homework than others.	T	57 (27.14)	11 (19.64)	46 (29.87)
19	Students with ADHD should be examined orally.	T	89 (42.38)	23 (41.07)	66 (42.86)
20	ADHD can be treated with only drugs.	F	46 (21.90)	12 (21.43)	34 (22.08)
21	ADHD is often caused by sugar intake and food additives.	F	125 (59.52)	41 (73.21)	84 (54.55)
22	ADHD is an issue of bad habits.	F	145 (69.05)	42 (75.00)	103 (66.88)
23	ADHD behavior is represented only within the school.	F	204 (97.14)	54 (96.43)	150 (97.40)
24	Excessive punishment increases behavioral problems in ADHD children.	T	174 (82.86)	46 (82.14)	128 (83.12)
25	ADHD is not a disease.	F	180 (85,71)	52 (92.86)	128 (83.12)

**Table 3 children-09-01775-t003:** Parents’ views on ADHD treatment and their concerns (N (%)).

Parents’ Attitudes and Beliefs	All Participants (n = 210)	Fathers (n = 56)	Mothers (n = 154)
Preferred treatment			
No need for treatment	1 (0.48)	0 (0.00)	1 (0.65)
Change habit	91 (43.33)	18 (32.14)	73 (47.40)
Behavior therapy	101 (48.10)	28 (50.00)	73 (47.40)
Medication	59 (28.10)	19 (33.93)	40 (25.97)
Psychological therapy	126 (60.00)	37 (66.07)	89 (57.79)
Traditional Chinese medicine treatment	22 (10.48)	6 (10.71)	16 (10.39)
Concerns about ADHD treatment			
Curative effect	133 (63.33)	34 (60.71)	99 (64.29)
Side effect	98 (46.67)	23 (41.07)	75 (48.70)
Treatment time	47 (22.38)	11 (19.64)	36 (23.38)
The aspects of the child the parents cared about the most			
Grade points	93 (44.29)	23 (41.07)	70 (45.45)
Peer relationship	56 (26.67)	15 (26.79)	41 (26.62)
Confidence	95 (45.24)	19 (33.93)	76 (49.35)
Psychological pressure	77 (36.67)	21 (37.50)	56 (36.36)
Personality growth	80 (38.10)	27 (48.21)	53 (34.42)
Health	69 (32.86)	25 (44.64)	44 (28.57)
Emotions	71 (33.81)	19 (33.93)	52 (33.77)

## Data Availability

Data described in this research are available from the corresponding author upon reasonable request.

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
