# Peer review of "Do Parents of Children with ADHD Know the Disease? Results from a Cross-Sectional Survey in Zhejiang, China"

_children, 2022, doi:10.3390/children9111775_

Round 1

Reviewer 1 Report

This interesting study explores if the parents of children with ADHD know different aspects of this condition. From the etiology to the type of management, academic and social relationships speciation among others.

The referenced bibliography is quite complete and additionally contains sources from different countries and continents.

The data obtained make possible to compare the kind of parent’s knowledge and highlights the importance of defining these aspects to establish support an education program for parents.

Author Response

This interesting study explores if the parents of children with ADHD know different aspects of this condition. From the etiology to the type of management, academic and social relationships speciation among others.

The referenced bibliography is quite complete and additionally contains sources from different countries and continents.

The data obtained make possible to compare the kind of parent’s knowledge and highlights the importance of defining these aspects to establish support an education program for parents.

Response: Thank you very much for your valuable comments on our MS,

Reviewer 2 Report

This study aims to evaluate the level of knowledge that parents in China have of ADHD. The writers correctly point out that parents' beliefs and understanding of ADHD likely influences how they made decisions about treatment for their children.  Educating parents about ADHD may therefore help promote appropriate treatment decisions.  The study aimed to assess the level of parents' knowledge of ADHD with a semi-structured questionnaire. 

The study design is appropriate and the findings add to the field.

I have two suggestions -

1. The literature review is inadequate.  

The paper should note that the home environment (which includes parents knowledge) is very important to child neurodevelopment. Cite Bush et al. 2022 Family environment, neurodevelopmental risk, and the environmental influences on child health outcomes (ECHO) initiative: Looking back and moving forward

The paper should also note that in addition to parental knowledge of ADHD, parental experiences, such as trauma, can influence their children's risk for psychiatric conditions such as ADHD.  Cite Duarte et al 2020 Intergenerational psychiatry: a new look at a powerful perspective

The paper should note that the scientific community is increasingly understanding how ADHD treatments work, which may help parents make treatment decisions.  Cite Wang et al. 2022 Causal effects of psychostimulants on neural connectivity: a mechanistic, randomized clinical trial

2. There are numerous idiomatic and grammatical errors.  There should be addressed.

Author Response

This study aims to evaluate the level of knowledge that parents in China have of ADHD. The writers correctly point out that parents' beliefs and understanding of ADHD likely influences how they made decisions about treatment for their children. Educating parents about ADHD may therefore help promote appropriate treatment decisions. The study aimed to assess the level of parents' knowledge of ADHD with a semi-structured questionnaire.

The study design is appropriate and the findings add to the field.

Response: Thank you for your comments.

I have two suggestions -

  1. The literature review is inadequate.

The paper should note that the home environment (which includes parents knowledge) is very important to child neurodevelopment. Cite Bush et al. 2022 Family environment, neurodevelopmental risk, and the environmental influences on child health outcomes (ECHO) initiative: Looking back and moving forward

The paper should also note that in addition to parental knowledge of ADHD, parental experiences, such as trauma, can influence their children's risk for psychiatric conditions such as ADHD. Cite Duarte et al 2020 Intergenerational psychiatry: a new look at a powerful perspective

The paper should note that the scientific community is increasingly understanding how ADHD treatments work, which may help parents make treatment decisions. Cite Wang et al. 2022Causal effects of psychostimulants on neural connectivity: a mechanistic, randomized clinical trial

Response: Many thanks for your valuable suggestions. We have added related information in the discussion section (Line 209-222, Line 304-313) and added 4 references (Reference 30, 31, 32, 44). Thank you again for your precious suggestions.

  1. There are numerous idiomatic and grammatical errors. There should be addressed.

Response: Many thanks for your valuable suggestions. We have found a native English-speaking expert to polish this MS. Thank you.

Reviewer 3 Report

Thank you for allowing me to review your paper. I recommend checking facts on item number 18, and item number 19, .table 2" Students with ADHD should do less homework" . The response to this question is marker "YES".  I believe students with ADHD should not do less home work instead ways should be incorporated to do same homework with psychotherapy or medications. similar comment for item number 19. 

Author Response

Thank you for allowing me to review your paper. I recommend checking facts on item number 18, and item number 19, .table 2" Students with ADHD should do less homework" . The response to this question is marker "YES". I believe students with ADHD should not do less home work instead ways should be incorporated to do same homework with psychotherapy or medications. similar comment for item number 19.

Response: Thank you for your comments. Our point is that children with ADHD should receive less homework than others and be examined orally. Most ADHD children have more problems in taking notes, completing family homework, school planning and learning motivation (1). We revised item 18 to clearly indicate that students with ADHD should do less homework than others and item 19 to indicate that students with ADHD should be examined orally. We are so sorry that our writing expression is not perfect to cause confusing. Thank you for your suggestion, and we revised the items in Table 2.

Reference:

(1). Calleja-Pérez B, Párraga JL, Albert J, et al. Trastorno por déficit de atención/hiperactividad. Hábitos de estudio [Attention deficit/hyperactivity disorder. Study habits]. Medicina (B Aires). 2019;79(Suppl 1):57-61.